# Effect of Empagliflozin with or without the Addition of Evolocumab on HDL Subspecies in Individuals with Type 2 Diabetes Mellitus: A Post Hoc Analysis of the EXCEED-BHS3 Trial

**DOI:** 10.3390/ijms25074108

**Published:** 2024-04-08

**Authors:** Isabella Bonilha, Érica Ivana Lázaro Gomes, Helison R. P. Carmo, Ikaro Breder, Joaquim Barreto, Jessica Breder, Daniel B. Munhoz, Luiz Sergio F. Carvalho, Thiago Quinaglia, Sheila T. Kimura-Medorima, Camila Moreira Gossi, Francesca Zimetti, Wilson Nadruz, Ilaria Zanotti, Andrei C. Sposito

**Affiliations:** 1Laboratory of Vascular Biology and Atherosclerosis (Aterolab), State University of Campinas (Unicamp), Sao Paulo 13083-887, Brazil; isaoliveira.ib@gmail.com (I.B.); ericai@unicamp.br (É.I.L.G.); helison.rafael@gmail.com (H.R.P.C.); ikarobreder@gmail.com (I.B.); joaquimbarretoantunes@gmail.com (J.B.); jecunha.silva@gmail.com (J.B.); dbmunhoz@gmail.com (D.B.M.); luizsergiofc@gmail.com (L.S.F.C.); tquinaglia@yahoo.com.br (T.Q.); sheilatk@gmail.com (S.T.K.-M.); camilams@unicamp.br (C.M.G.); 2Cardiovascular Center Aalst, OLV Clinic, 9300 Aalst, Belgium; 3Department of Food and Drug, University of Parma, 43124 Parma, Italy; francesca.zimetti@unipr.it (F.Z.); ilaria.zanotti@unipr.it (I.Z.); 4Division of Cardiology, State University of Campinas (Unicamp), Sao Paulo 13083-887, Brazil; wilnj@unicamp.br

**Keywords:** PCSK9i, high-density lipoprotein, empagliflozin, evolocumab, type 2 diabetes mellitus

## Abstract

Evolocumab and empagliflozin yield a modest rise in plasma high-density lipoprotein cholesterol (HDL-C) through unknown mechanisms. This study aims to assess the effect of evolocumab plus empagliflozin vs. empagliflozin alone on HDL subspecies isolated from individuals with type 2 diabetes mellitus (T2D). This post hoc prespecified analysis of the EXCEED-BHS3 trial compared the effects of a 16-week therapy with empagliflozin (E) alone or in combination with evolocumab (EE) on the lipid profile and cholesterol content in HDL subspecies in individuals with T2D divided equally into two groups of 55 patients. Both treatments modestly increased HDL-C. The cholesterol content in HDL subspecies 2a (7.3%), 3a (7.2%) and 3c (15%) increased from baseline in the E group, while the EE group presented an increase from baseline in 3a (9.3%), 3b (16%) and 3c (25%). The increase in HDL 3b and 3c was higher in the EE group when compared to the E group (*p* < 0.05). No significant interactive association was observed between changes in hematocrit and HDL-C levels after treatment. Over a 16-week period, empagliflozin with or without the addition of evolocumab led to a modest but significant increase in HDL-C. The rise in smaller-sized HDL particles was heterogeneous amongst the treatment combinations.

## 1. Introduction

Type 2 diabetes mellitus (T2D) is a critical global public health challenge as it raises the risk of all-cause mortality up to 15%, primarily due to cardiovascular complications [1]. Although the pathophysiology of T2D is complex, the non-enzymatic glycation reaction is one of the main factors in the progression and complications induced by diabetes [2]. Insulin resistance (IR), the origin of T2D, is intricately linked with atherogenic dyslipidemia, affecting triglyceride metabolism and both low-density lipoprotein (LDL) and high-density lipoprotein cholesterol (HDL-C) [3,4].

In individuals with T2D, lowering LDL cholesterol (LDL-C) is a key strategy to prevent atherosclerotic cardiovascular events. The development of monoclonal antibodies against proprotein convertase subtilisin/kexin type 9 (PCSK9) has enabled additional potent reductions in LDL-C and, consequently, reduction in cardiovascular risk in patients with and without diabetes [5]. In individuals with IR, with or without T2D, plasma PCSK9 concentration increases are directly correlated to cholesteryl ester transfer protein (CETP) activity increase [6]. Consequently, the size and cholesterol content of HDL particles is reduced, which also happens due to synthesis and reduced removal of triglyceride-rich lipoproteins. This may potentially explain the modest increase in HDL-C levels observed after PCSK9 inhibitor (PCSK9i) therapy [5,7,8,9]. If this holds true, by inhibiting PCSK9 and reducing CETP activity, the increase in HDL-C should primarily result from an increase in large HDL particles.

Empagliflozin, a sodium-glucose co-transporter 2 (SGLT2) inhibitor, has demonstrated benefits in reducing cardiovascular events and mortality in T2D patients [10]. Additionally, empagliflozin increases HDL-C and apolipoprotein A-I in direct proportion to surrogate markers of hemoconcentration, such as hematocrit and serum albumin [11]. However, triglyceride changes after empagliflozin happen in the opposite direction, suggesting that a change in metabolic behavior may also contribute to the lipid profile change [11]. Differently than the hemoconcentration hypothesis, enhancing triglyceride metabolism is expected not only to elevate HDL-C concentration but also to increase the size of HDL particles. Nevertheless, data to support this hypothesis are currently limited.

This study is a prespecified subanalysis of the EXCEED-BHS3 trial and aims to evaluate the effects of empagliflozin alone (E) or combined with evolocumab (EE) on the plasma concentration of HDL subspecies (2b, 2a, 3a, 3b, and 3c) in individuals with T2D. Understanding the effect of these agents on HDL phenotype may provide valuable insights into their mechanisms of action and, consequently, their cardiovascular benefits in T2D.

## 2. Results

A total of 110 participants met the inclusion criteria and underwent 1:1 randomization into either E or EE groups. Demographic characteristics were previously detailed in [12]. In summary, the mean age was 58 years, the mean T2D duration was 9 years, and 71% of patients were male. Except for higher HDL-C pretreatment in the EE group, baseline characteristics were comparable (Table 1).

After 16 weeks of treatment, changes in systolic blood pressure, diastolic blood pressure and mean reduction in HbA1c and fasting glucose were equivalent between groups, as seen in Table 2.

Table 3 shows the cholesterol content of plasma lipoproteins, including the HDL subspecies, at baseline and post-therapy. After 16 weeks of treatment, both the E and EE groups exhibited significant reductions in very-low-density lipoprotein cholesterol (VLDL-C) and triglyceride (TG) levels and modestly increase in HDL-C levels. Furthermore, as predicted, the EE group demonstrated greater LDL-C reductions compared to the E group (Table 3).

Intergroup analysis also reveals increased concentrations of HDL subspecies 3a, 3b and 3c in group EE and subspecies 2a in group E (Table 3). Likewise, the proportion of each HDL subspecies was also altered in the EE group, increasing the percentage change of HDL subspecies 3b and 3c in relation to total HDL (Figure 1A).

In the intragroup analysis of group E, a notable increase in cholesterol content in HDL subspecies 2a, 3a and 3c was observed after 16 weeks of treatment, as described in Table 3. On the other hand, the treatment had no discernible impact on the percentage change of each HDL subspecies, as depicted in Figure 1B. Within the EE group, HDL subfractions 3a, 3b and 3c increased, as evidenced by the data in Table 3 and illustrated in Figure 1C.

Both groups exhibited hematocrit increase after 16 weeks of treatment. Group E had a 2.6% rise (from 41 ± 4% to 43 ± 4%; *p* < 0.001), while Group EE experienced a 1.6% increase (from 40 ± 4% to 42 ± 6; *p* = 0.032). No association was found between changes in hematocrit and changes in the concentration or proportion of total HDL-C in the population (Figure 2A). Mediation analysis was conducted to examine the indirect effect of change in hematocrit (mediating variable) on the relationship between the treatment group and change in HDL concentration. However, the results did not indicate a statistically significant indirect effect (*p* > 0.05). Thus, the data do not provide convincing statistical evidence that the change in hematocrit mediated the relationship between treatment and the change in HDL concentration (Figure 2B).

## 3. Discussion

Among the effective treatments for reducing the risk of cardiovascular events in individuals with T2D, SGLT2 inhibitors (SGLT2i) and PCSK9i emerge prominently. Initial data indicated an elevation in HDL-C with both therapies, attributed to hemoconcentration following SGLT2i [13] and increased CETP activity after PCSK9i [6]. However, according to Kashyap et al. [14], HDL levels considered normal or even high, do not equate to adequate functionality in diabetic individuals. Therefore, to clarify the effects of both therapies, the impact of a 16-week regimen of these therapies on absolute values as well as the distribution pattern of HDL subspecies was systematically evaluated in this post hoc analysis of the phase 2 translational study EXCEED-BHS3 trial.

In contrast to the supposed link between HDL-C increase and hemoconcentration, our study, for the first time, reveals an absence of relationship between changes in HDL-C and hematocrit after SGLT2i therapy. This lack of correlation challenges the proposed hypothesis of hemoconcentration. In addition, we identified a nonuniform rise across HDL subspecies, resulting in a shift favoring smaller-sized particles. Notably, no alterations in CETP or PCSK9 levels have been previously reported following SGLT2i treatment [12,15], limiting the likelihood of an impact on this pathway. The lack of changes in these enzymes reduces the possibility that they play an important role or are affected by SGLT2i treatment. Studies have shown that SGLT2i treatment led to a significant increase in HDL-C, while triglyceride levels were reduced [16]. The proposed mechanisms of action are related to the improvement of insulin sensitivity and insulin secretion, leading to a reduction in hepatic synthesis and an increase in the catabolism of lipoproteins rich in triglycerides [17]. Thus far, the underlying mechanisms of how SGLT2i increases HDL-C concentration and modify HDL subspecies distribution remain elusive.

Consistent with prior research, we noted a modest yet significant increase in HDL-C following evolocumab treatment in combination therapy [8,18,19]. Furthermore, our findings align with existing studies [20], indicating that this treatment promotes an increase in smaller HDL subspecies. Beyond the previously documented impact of PCSK9i on CETP activity, the notable reduction in apolipoprotein B-containing lipoproteins, such as LDL-C and VLDL-C, likely contributed to the observed HDL remodeling—a phenomenon observed in previous statin studies [21]. In fact, these lipoproteins serve as a triglyceride source for HDL cholesterol exchange through CETP, directly influencing HDL particle remodeling [22]. While our analysis did not explore the detailed mechanisms of formation underlying HDL subspecies remodeling, it is plausible that the effect of PCSK9i on HDL particles is strongly influenced by the change in lipid transfer between apolipoprotein A-containing and apolipoprotein B-containing lipoproteins.

This study has inherent limitations that warrant consideration. Firstly, being a secondary analysis of a phase 2 translational study, the analyses should be viewed as exploratory and hypothesis-generating. Hence, the results are not meant to provide guidance for clinical practice, but rather to shed light on potential mechanisms of benefit from the experimental therapies. Secondly, due to the nature of the study design, it is not feasible to isolate the effect of evolocumab alone; rather, it reflects its combination with empagliflozin. Despite these constraints, the study unveils original findings that contribute to a deeper understanding of the comprehensive metabolic effects of the two therapeutic classes under investigation.

## 4. Materials and Methods

### 4.1. Study Design

The present study is a prespecified post hoc analysis of the Expanded Combination of Evolocumab plus Empagliflozin on Diabetes Trial (EXCEED-BHS3 Trial), which was a phase 2 translational, prospective, randomized, single-center, and open-label study [23]. This prespecified analysis focuses primarily on surrogate endpoints related to HDL subspecies distribution and not on assessments of clinical hard endpoints. The study is registered on clinicaltrials.gov under the code NCT03932721 and received approval from the institutional ethics committee, as registered with CAAE: 88800718.0.0000.5404. As previously described [23], the research involved the participation of individuals with T2D within the age range of 40 to 70 years and with a body mass index (BMI) below 40 kg/m^2^. During the screening phase, the following parameters were evaluated: (a) glycated hemoglobin between 7 and 9%; (b) LDL-C levels between 70 and 100 mg/dL, using the maximum tolerated dose of simvastatin or rosuvastatin; (c) blood pressure (BP) less than 140/90 mmHg; and (d) flow-mediated dilation (FMD) between 1 and 12%. After screening and necessary adjustments, participants who met these criteria were randomized to a 16-week treatment, which consisted of daily administration of empagliflozin (25 mg) or the combination of empagliflozin (25 mg) and evolocumab (140 mg per day) every two weeks. Throughout the 16 weeks of treatment, at each visit endothelial function was assessed and blood samples were collected for biochemical analysis.

### 4.2. HDL Subspecies Isolation and Cholesterol Dosage

According to the density of HDL, two classes are identified: (i) large, light HDL2, whose densities vary from 1.063–1.125 g/mL, and (ii) small, dense HDL3 with a density of 1.0125–1.21 g/mL [24]. HDL subspecies were isolated from 3 mL of plasma adjusted with potassium bromide (KBr) to a density of 1.21 g/mL. A total of 2 mL of saline solution with a density of 1.24 g/mL was added into ultra-clear tubes (Beckman, Brea, CA, USA), carefully forming a density gradient with the aid of a mini peristaltic pump (Thomas Scientific, Swedesboro, NJ, USA), followed by 3 mL of plasma adjusted to 1.21 g/mL, 2 mL of saline solution with a density of 1.063 g/mL, 2.5 mL of saline solution with a density of 1.019 g/mL and, finally, 3 mL of saline solution with a density of 1.006 g/mL. The density of the solutions was verified using a densitymeter, Easy D40 (Mettler Toledo, Sao Paulo, Brazil). The gradient formed was ultracentrifuged at 40,000 rpm 15 °C for 48 h in a SW41-Ti rotor in a Beckman L8-80M ultracentrifuge (Brea, CA, USA). After ultracentrifugation, the non-HDL lipoprotein fractions were discarded (approximately 5 mL) and the HDL subspecies were extracted in the sequence 700 µL of 2b (d 1.063–1.087 g/mL), 800 µL of 2a (d 1.088–1.110 g/mL), 800 μL of 3a (d 1.110–1.129 g/mL), 800 μL of 3b (d 1.129–1.154 g/mL) and 800 µL of 3c (d 1.154–1.170 g/mL). At the end of the ultracentrifugation, total cholesterol was measured in the five HDL subspecies with the SX-140 biochemical analyzer (Sinnowa, Sao Paulo, Brazil), using a commercially available kit (Biotecnica, Minas Gerais, Brazil).

### 4.3. Statistical Analysis

The normality of quantitative variables was verified using the Shapiro–Wilks test. Continuous data were expressed as median and in the interquartile range (IQR) if non-parametric, whereas normally distributed data were presented as mean ± standard deviation. The analyses of the change in HDL subspecies between groups were performed using the covariance analysis (ANCOVA), adjusting for relative basal total HDL concentrations. For intragroup analyses, the paired *t*-test was performed. Statistical analysis was conducted using SPSS software, version 25.0 (SPSS Inc., Chicago, IL, USA). A path analysis was used to assess a possible mechanism through which HDL changed from baseline (at the randomization visit) to last visit (at 16 weeks). The analysis was performed through structural equation modeling (Delta method and resampling method by percentile bootstrapping).

## 5. Conclusions

In conclusion, our post hoc analysis of the EXCEED-BHS3 trial sheds light on the effects of SGLT2i and PCSK9i on HDL particles from individuals with T2D. Our study confirms a modest albeit significant increase in HDL-C and unveils a novel nonuniform rise across HDL subspecies, favoring particles of intermediate and small sizes, such as 2a, 3a, 3b and 3c in both therapies. While our study did not capture clinical events that would be correlated with the changes in HDL subspecies distribution, this analysis serves as a foundation for formulating hypotheses that may warrant further investigation through extensive, long-term clinical trials focusing on robust endpoints.

## Figures and Tables

**Figure 1 ijms-25-04108-f001:**
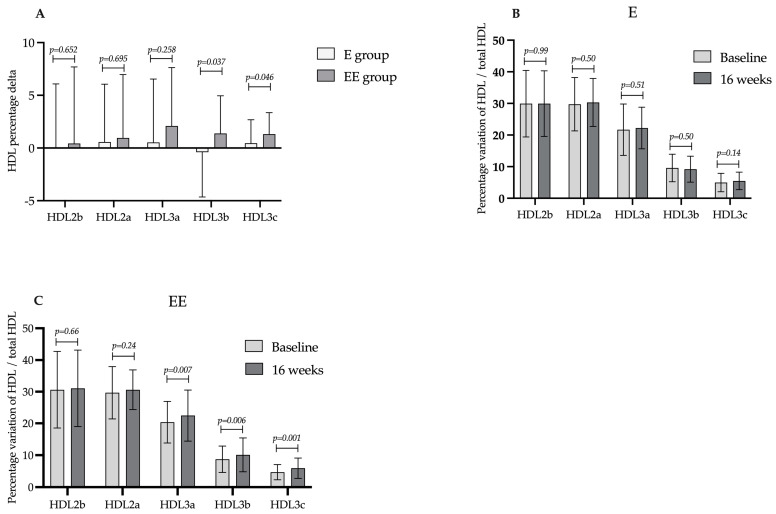
Percentage of each HDL subspecies variation between treatments in relation to total HDL. Data presented as median ± standard deviation. (**A**), intergroup analysis of the percentage change in HDL subspecies after 16 weeks of treatment. (**B**,**C**), intragroup analysis of the percentage variation of each HDL subspecies in relation to total HDL after 16 weeks of treatment in group E and group EE, respectively. This proportion is based on the total cholesterol content measured.

**Figure 2 ijms-25-04108-f002:**
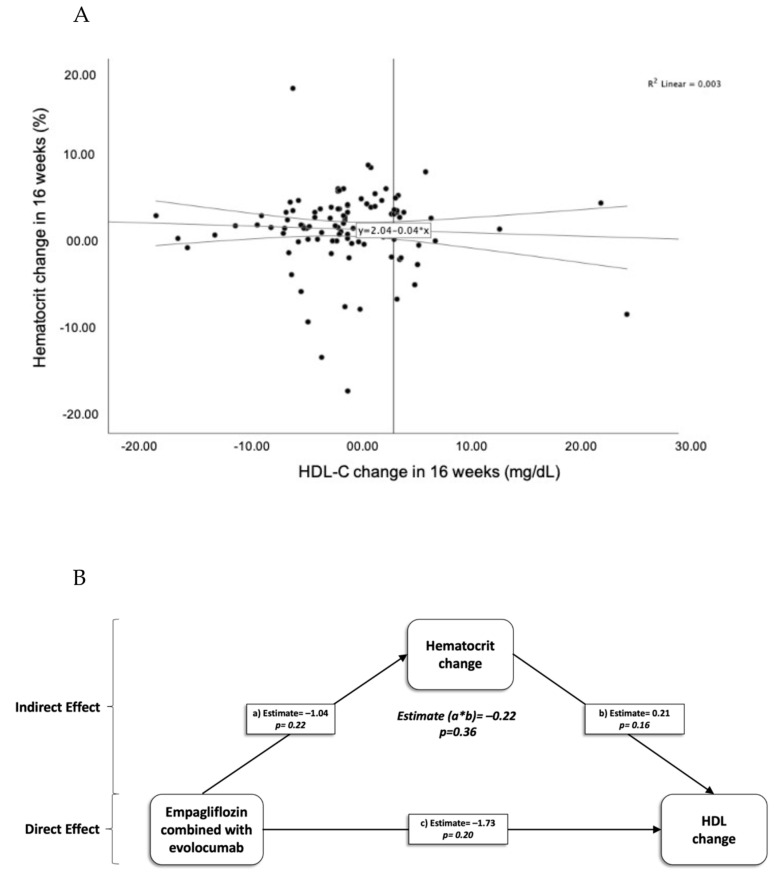
(**A**) Analysis of the association between hematocrit and changes in HDL-C after 16 weeks of treatment in the population. (**B**) Mediation effect of hematocrit on HDL concentration in relation to treatment.

**Table 1 ijms-25-04108-t001:** Baseline demographic characteristics of the study population.

	E	EE
	55 Patients	55 Patients
Gender male, %	69	73
Age, years	58 ± 8	58 ± 6
Body Mass Index, kg/m^2^	31 ± 4.2	31 ± 5.1
Waist circumference, cm	106 ± 15	105 ± 14
Hypertension, %	67	76
T2D duration, years	8 (5–15)	10 (6–15)
HbA1c, %	8.0 (7.5–8.3)	7.7(7.2–8.4)
HDL-C, mg/dL	36 (32–43)	41 (35–47) *
LDL-C, mg/dL	83 ± 13	84 ± 13
VLDL-C, mg/dL	32 (25–44)	33 (22–43)
Triglycerides, mg/dL	166 (110)	163 (108)
Thiazides diuretics, %	33	38
Oral hypoglycemic, %	44	38
Sulfonylureas, %	44	45
Metformin, %	100	100
Statin use, %	100	100

HbA1c, glycated hemoglobin; LDL-C, low-density lipoprotein cholesterol; HDL-C, high-density lipoprotein cholesterol; VLDL-C, very-low-density lipoprotein cholesterol. *, *p* < 0.05.

**Table 2 ijms-25-04108-t002:** Post-treatment changes at 16 weeks.

Variable	E	EE	*p*
Office SBP, mmHg	−10 (18)	−4 (20)	0.284
Office DBP, mmHg	4 (11)	−2 (13)	0.197
Fasting Blood Glucose, mg/dL	−38 (48)	−41 (44.5)	0.089
HbA1c, %	−0.7 (1.0)	−0.9 (0.9)	0.824

SBP systolic blood pressure, DBP diastolic blood pressure, HbA1c glycosylated hemoglobin.

**Table 3 ijms-25-04108-t003:** Intragroup and intergroup analysis of lipid profile and cholesterol concentration in HDL subspecies after 16 weeks of treatment.

	E				EE				
	Baseline	16 Weeks	*p* ^†^	Percentage Changes	Baseline	16 Weeks	*p* ^†^	Percentage Changes	*p* ^#^
HDL-C	36 ± 9	39 ± 10	0.004 *	8.3%	40 ± 11	42 ± 11	<0.001 *	5.0%	0.430
LDL-C	71 ± 18	70 ± 13	0.063	0	71 ± 18	28 ± 16	<0.001 *	−60.5%	<0.001 *
VLDL-C	36 ± 15	31 ± 15	0.008 *	−13.8%	34 ± 15	26 ± 12	<0.001 *	−23.5%	0.042 *
TG	179 ± 77	154 ± 74	0.008 *	−13.9%	170 ± 75	129 ± 62	<0.001 *	−24.1%	0.042 *
HDL2b-C	11.4 ± 5.1	12.1 ± 5.4	0.068	6.1%	13.1 ± 6.9	13.4 ± 7.2	0.528	2.3%	0.247
HDL2a-C	10.9 ± 3.3	11.7 ± 3.1	0.020 *	7.3%	12.5 ± 3.9	13.0 ± 3.6	0.155	4.0%	<0.001 *
HDL3a-C	8.3 ± 3.6	8.9 ± 3.3	0.040 *	7.2%	8.6 ± 3.1	9.4 ± 3.4	0.009 *	9.3%	<0.001 *
HDL3b-C	3.8 ± 2.2	3.8 ± 2.0	0.951	0	3.6 ± 1.8	4.2 ± 2.4	0.011 *	16.6%	<0.001 *
HDL3c-C	2.0 ± 1.4	2.3 ± 1.3	0.043 *	15.0%	2.0 ± 1.1	2.5 ± 1.3	<0.001 *	25.0%	0.002 *

Data presented as mean ± standard deviation of total cholesterol concentrations in mg/dL. ^†^ Intragroup analyses; ^#^ intergroup analyses. Percentage changes show intragroup variation. The intergroup variation is demonstrated in the graphs in Figure 1. HDL-C, high-density lipoprotein cholesterol; LDL-C, low-density lipoprotein cholesterol; VLDL, very-low-density lipoprotein cholesterol; TG, triglycerides. *, statistically significant differences.

## Data Availability

The data presented in this study are available on request from the corresponding author.

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
