# Peer review of "Effect of Empagliflozin with or without the Addition of Evolocumab on HDL Subspecies in Individuals with Type 2 Diabetes Mellitus: A Post Hoc Analysis of the EXCEED-BHS3 Trial"

_ijms, 2024, doi:10.3390/ijms25074108_

Round 1
Reviewer 1 Report
Comments and Suggestions for Authors
The Manuscript entitled "Effect of Empagliflozin combined or not with Evolocumab on 2 HDL subspecies in individuals with type 2 Diabetes Mellitus: a 3 posthoc analysis of the EXCEED-BHS3 trial" talks about the evolocumab and empagliflozin vs empagliflozin alone on HDL subspecies isolated from 23 type 2 diabetes mellitus (T2D) individuals. This post-hoc analysis of a 16-week therapy with empagliflozin (E) alone or in combination with 25 evolocumab (EE) on the lipid profile and cholesterol content in HDL subspecies in individuals with T2D of two groups of 55 patients. Both treatments modestly increased HDL-C. The cholesterol content in HDL subspecies 2a, 3a and 3c increased slightly in the E group, while the EE group the same subspecies increased more than E group alone. The Authors claims that a 16-week period, empagliflozin combined or not with evolocumab led to a modest but significant increase in HDL-C.
This reviewer has certain which needs to replied on point by point basis-
1. Use of scientific english is suggested instead of non-scientific, for example at one instance Authors have used 'solo'. Kindly proof read for whole manuscript for non-scientific words.
2. How the authors came to the conclusion for doing the 16 week trials, kindly give reference where it is mentioned.
3. Authors have proposed that EE have profound impact on HDLC 3a, 3b and 3 c are enhanced significantly as can be seen in the figure 1C. Did Authors performed any toxicity studies as well for the concentrations of E & E?
4. In the Introduction Authors did not mention the major cause for diabetes type 2 which leads to CVDs. Authors are advised to mention with proper citations-
Non-enzymatic glycation reaction is one of the major factor for the development of diabetes type 2 [ref. A Perspective on the Impact of Advanced Glycation End Products in the Progression of Diabetic Nephropathy. Current Protein and Peptide Science. 2023;24(1):2-6.].
5. In discussion Authors claimed, for the first time, reveals an absence of relationship between changes in HDL- C and hematocrit after SGLT2i therapy. How did the author came to this statement? discuss.
6. In Conclusion authors should also mention the subspecies of HDL which is important in context to this study.
Comments on the Quality of English LanguageEnglish is okay.
Author Response
The Manuscript entitled "Effect of Empagliflozin combined or not with Evolocumab on HDL subspecies in individuals with type 2 Diabetes Mellitus: a posthoc analysis of the EXCEED-BHS3 trial" talks about the evolocumab and empagliflozin vs empagliflozin alone on HDL subspecies isolated from type 2 diabetes mellitus (T2D) individuals. This post-hoc analysis of a 16-week therapy with empagliflozin (E) alone or in combination with evolocumab (EE) on the lipid profile and cholesterol content in HDL subspecies in individuals with T2D of two groups of 55 patients. Both treatments modestly increased HDL-C.  The cholesterol content in HDL subspecies 2a, 3a and 3c increased slightly in the E group, while the EE group the same subspecies increased more than E group alone. The Authors claims that a 16-week period, empagliflozin combined or not with evolocumab led to a modest but significant increase in HDL-C. 
This reviewer has certain which needs to replied on point by point basis:
- Use of scientific english is suggested instead of non-scientific, for example at one instance Authors have used 'solo'. Kindly proof read for whole manuscript for non-scientific words.
Response from authors: We thank the reviewer for the helpful suggestion. We took your suggestion into consideration and changed the word 'solo' to 'alone', as evidenced by the yellow stripe marking in the manuscript.
- How the authors came to the conclusion for doing the 16 week trials, kindly give reference where it is mentioned.
Response from authors: We thank the reviewer for the pertinent question raised. As reported in our seminal work (10.1177/2040622320959248), we chose a 16-week treatment, considering the primary endpoint, which was the change in 1-minute flow-mediated dilation (FMD) from baseline to 16 weeks of treatment. This choice was motivated by the fact that endothelial function is capable of quickly adapting to changes, especially through dilation and stimulation of nitric oxide production, as demonstrated in previous studies. Furthermore, we observed that this period is adequate to detect significant changes in the lipid profile.
- Authors have proposed that EE have profound impact on HDLC 3a, 3b and 3 c are enhanced significantly as can be seen in the figure 1C. Did Authors performed any toxicity studies as well for the concentrations of E & E?
Response from authors: We appreciate your valuable comment. Although we recognize the importance of evaluating the toxicity of therapeutic agents, our study focused primarily on the effects of these medications on cholesterol concentrations, especially HDL subfractions. It is important to highlight that empagliflozin has positive pharmacokinetic properties, such as rapid absorption and renal elimination (10.1007/s40262-013-0126-x). Furthermore, evolocumab has been shown to be safe and well tolerated, and its addition to background lipid-lowering therapy is not associated with an increased risk of adverse events or toxicity (10.1155/2023/7362551). In our study, we monitored the incidence of adverse events (AEs) every 15 days, through telephone contact or during in-person consultations. However, regrettably, we do not conduct specific trials to assess toxicity.
- In the Introduction Authors did not mention the major cause for diabetes type 2 which leads to CVDs. Authors are advised to mention with proper citations-
Non-enzymatic glycation reaction is one of the major factor for the development of diabetes type 2 [ref.  A Perspective on the Impact of Advanced Glycation End Products in the Progression of Diabetic Nephropathy. Current Protein and Peptide Science. 2023;24(1):2-6.].
Response from authors: We thank the reviewer for the excellent suggestion. We have added a description in the introduction about the cause that leads to cardiovascular disease in type 2 diabetes mellitus and included the recommended reference. From line 41 to 42, as follows: “Although the pathophysiology of T2D is complex, the non-enzymatic glycation reaction is one of the main factors in the progression and complications induced by diabetes”.
- In discussion Authors claimed, for the first time, reveals an absence of relationship between changes in HDL- C and hematocrit after SGLT2i therapy. How did the author came to this statement? discuss.
Response from authors: We thank you for your insightful review. In the discussion of our study, we stated that, for the first time, we revealed the absence of a relationship between changes in HDL-C and hematocrit after SGLT2i therapy. This statement was based on some observations and analyzes carried out. We referenced a previous study that showed an increase in hematocrit associated with an increase in lipid fractions with the use of empagliflozin. This increase was attributed to hemoconcentration resulting from the increase in urinary volume observed. In our analysis, we observed an increase in hematocrit after 16 weeks of treatment in both groups. This increase was consistent with previous observations and led us to further examine the relationship between this change in hematocrit and changes in HDL-C levels.
Therefore, we performed an initial association analysis between the change in hematocrit and the change in HDL-C concentration, in which we did not observe a statistically significant effect. Given this, we decided to perform a mediation analysis to examine the indirect effect of change in hematocrit on the relationship between treatment group and change in HDL-C concentration. However, the results of the mediation analysis did not find a statistically significant connection between the change in hematocrit and the change in HDL-C levels. Despite of this, we agreed that this expression would induce an expectation of solving or originating a new data and we removed “for the first time” from the text.
- In Conclusion authors should also mention the subspecies of HDL which is important in context to this study.
Response from authors: We thank the reviewer for the suggestion. We added in the conclusion which subspecies increased. From line 229 to 230, as follows: "favoring particles of intermediate and small sizes, such as 2a, 3a, 3b and 3c in both therapies".
Reviewer 2 Report
Comments and Suggestions for Authors
In this study, the authors found that both E and EE treatments modestly increased HDL-C. The cholesterol content in HDL subspecies 2a (7.3%), 3a (7.2%), and 3c (15%) increased from baseline in the E group, while the EE group presented an increase from baseline in 3a (9.3%), 3b (16%), and 3c (25%). The increase in HDL 3b and 3c was higher in the EE group when compared to the E group(p<0.05). The authors should clarify the significance of the analysis of different subtypes of high-density lipoprotein in patients with type 2 diabetes. Why E or EE treatment caused changes in different subtypes of high-density lipoprotein in patients with type 2 diabetes? What do the changes in different subspecies of high-density lipoprotein caused by two treatments indicate?
Why do the authors only focus on the indicator of high-density lipoprotein subspecies? There are many indicators related to diabetes. Are there any differences between the two treatments?
The authors need to further discuss the reasons for the differences in the two treatments from a mechanistic perspective.
Comments on the Quality of English LanguageNo
Author Response
Response from authors: We appreciate the reviewer’s relevant question. Patients with T2DM have an increased risk of cardiovascular disease. HDL is a heterogeneous class of particles, which vary in size, density, composition and function. However, not all HDL particles are equal in terms of their ability to protect against cardiovascular disease. According to Kashyap, levels considered normal or even high of HDL in diabetic individuals are not equivalent to their adequate functionality (10.1210/jc.2017-01551). Studies have revealed significant changes in HDL size in diabetic individuals, with loss of large and very large HDL2 and gain of small HDL3, rich in TGs and poor in cholesterol (10.1016/S1056-8727(01)00159-3). Therefore, understanding the distribution and characteristics of different HDL subtypes can provide important information about cardiovascular risk in patients with T2DM. We added this information in the text.
Studies have shown that SGLT2i treatment led to a significant increase in HDL-C, while triglyceride levels were reduced, making SGLT2i highly useful for the clinical treatment of dyslipidemia in diabetic patients (10.1016/j.phrs.2020.105068). The proposed mechanisms of action are related to the improvement of insulin sensitivity and insulin secretion, leading to reduced hepatic synthesis and increased catabolism of triglyceride-rich lipoproteins (10.1080/17425255.2018.1541348). We added the proposed mechanism of action using SGLT2i to the discussion, highlighted in yellow.
Regarding PCSK9 monoclonal antibodies like evolocumab, although several data have been reported on their LDL-C lowering effect, studies show a modest increasing effect on HDL-C levels (10.1056/NEJMoa1615664). The decrease in triglyceride-rich lipoprotein concentrations following administration of PCSK9i leads to a reduction in CETP activity, reducing lipid exchange between particles of HDL and TG-rich lipoproteins. Although our analysis did not investigate in detail the mechanisms underlying the modulation of HDL subspecies, it is plausible to postulate that this modulation is associated with a decrease in CE transfer from HDL to apoB-rich lipoproteins, thus promoting increased CE-rich HDL particles.
Why do the authors only focus on the indicator of high-density lipoprotein subspecies? There are many indicators related to diabetes. Are there any differences between the two treatments?
Response from authors: We thank the reviewer for the comments. In this subanalysis we focused only on HDL subfractions, however, we showed in a previous publication that in patients receiving SGLT2i, the addition of evolocumab improves flow-mediated arterial dilation and nitric oxide production both under resting conditions and after stimulation with ischemia and reperfusion (10.1186/s12933-022-01584-8). And we are working on another production that evaluates the concentration of cholesterol in the five subfractions of LDL and the lipidomics carried out at the beginning and after 16 weeks in both treatments.
The authors need to further discuss the reasons for the differences in the two treatments from a mechanistic perspective.
Response from authors: We thank the reviewer for the suggestion. The mechanisms are not defined, and more studies are needed to propose the pathways that mediate these changes. However, we have added proposed mechanisms. From line 151 to 155, as follows: “Studies suggest that the mechanisms of action are related to the improvement of insulin sensitivity and insulin secretion, leading to a reduction in hepatic synthesis and an increase in the catabolism of lipoproteins rich in triglycerides.” Regarding PCSK9i, the proposed mechanism is under the activity of CETP. As there is a very significant decrease in rich lipoproteins containing apoB, the exchange of triglycerides with HDL cholesterol is influenced, promoting particle remodeling. Proposed mechanism from line 157 to 165.
Reviewer 3 Report
Comments and Suggestions for Authors
In their interesting paper, the authors describe the effects of either empagliflozin alone or in combination with evolocumab on different HDL subspecies. While the main CV benefit of empagliflozin does not pertain to lipids at all and the benefit of evolocumab is mediated by dramatic LDL-C lowering, described effects on HDL particles are of interest to a lipidologist. Interestingly, the authors demonstrated that effects of empagliflozin do not depend on hemoconcentration, leaving the MOA affecting HDL particles an open question. Effects of evolocumab on lipoprotein species have been known before, so there were no surprises here. The manuscript is well written and supported by appropriate citations. My only minor suggestion is to include in Tab. 2 percentage changes for all lipid parameters in both E and EE groups, in addition to the absolute values.
Author Response
In their interesting paper, the authors describe the effects of either empagliflozin alone or in combination with evolocumab on different HDL subspecies. While the main CV benefit of empagliflozin does not pertain to lipids at all and the benefit of evolocumab is mediated by dramatic LDL-C lowering, described effects on HDL particles are of interest to a lipidologist. Interestingly, the authors demonstrated that effects of empagliflozin do not depend on hemoconcentration, leaving the MOA affecting HDL particles an open question. Effects of evolocumab on lipoprotein species have been known before, so there were no surprises here. The manuscript is well written and supported by appropriate citations. My only minor suggestion is to include in Tab. 2 percentage changes for all lipid parameters in both E and EE groups, in addition to the absolute values.
Response from authors: We thank the reviewer for reading the manuscript and for the suggestion. We added two columns to table 2 that show the intragroup percentage changes. We also include it in the legend, from line 96 to 97, as follows: Percentage changes show intragroup variation. The intergroup variation is demonstrated in the graphs in figure 1.
Reviewer 4 Report
Comments and Suggestions for Authors
In this paper, the authors evaluated the effect of evolocumab plus empagliflozin vs empagliflozin alone on HDL subspecies isolated from people with type 2 diabetes (T2D). Their analysis involves comparing the effects of a 16-week treatment with empagliflozin alone or in combination with evolocumab (EE). The lipid-lowering effect of evolocumab in the subgroup of T2DM patients with low HDL-C has been previously reported, but the effects of empagliflozin or combined with evolocumab on the plasma concentration of HDL subspecies (2b, 2a, 3a, 3b, and 3c) may provide valuable insights into their mechanisms of action.
I think it is a relevant and well-structured article.
I recommend the authors to delete the column from table 1 with p. It is enough to mention if it is greater or less than 0.5.
I want to ask the authors why they chose to evaluate the patients for a period of 16 weeks? Is there any reason?
I suggest the authors mention in the conclusions which subspecies increased.
Author Response
In this paper, the authors evaluated the effect of evolocumab plus empagliflozin vs empagliflozin alone on HDL subspecies isolated from people with type 2 diabetes (T2D). Their analysis involves comparing the effects of a 16-week treatment with empagliflozin alone or in combination with evolocumab (EE). The lipid-lowering effect of evolocumab in the subgroup of T2DM patients with low HDL-C has been previously reported, but the effects of empagliflozin or combined with evolocumab on the plasma concentration of HDL subspecies (2b, 2a, 3a, 3b, and 3c) may provide valuable insights into their mechanisms of action.
I think it is a relevant and well-structured article.
I recommend the authors to delete the column from table 1 with p. It is enough to mention if it is greater or less than 0.5.
Response from authors: We thank the reviewer for the great suggestion. We made changes to table 1, removing the column with the p value, as suggested by the reviewer.
I want to ask the authors why they chose to evaluate the patients for a period of 16 weeks? Is there any reason?
Response from authors: We appreciate the reviewer’s relevant question. As reported in our seminal work (10.1177/2040622320959248), the choice of the 16-week treatment duration considered the primary endpoint, which consisted of evaluating the change in 1-minute flow-mediated dilation (FMD) from baseline to 16 weeks of treatment. This decision was based on the rapid ability to adapt endothelial function to changes, especially due to dilation and stimulation of nitric oxide production, as documented in previous studies. Furthermore, we emphasize that this period is sufficient to identify significant changes in the lipid profile.
I suggest the authors mention in the conclusions which subspecies increased.
Response from authors: We thank the reviewer for the suggestion. We added in the conclusion which subspecies increased. From line 229 to 230, as follows: "favoring particles of intermediate and small sizes, such as 2a, 3a, 3b and 3c in both therapies".
Reviewer 5 Report
Comments and Suggestions for Authors
In this study, Bonilha et al. analyzed data of the EXCEED-BHS3 trial to seek the therapeutic effects of the combination of empagliflozin and evolocumab in patients with type 2 diabetes. This clinical trial is registered as NCT03932721, and the authors published a paper to show its concepts and methods (ref#19). The authors published another paper to show that plasma levels of nitrate and nitrite were increased in EE group, compared to E group (ref#11). In this manuscript, the authors showed that serum levels of HDL-C were significantly elevated in both E and EE groups, and the effects of EE were greater than E only. However, NCT03932721 is a phase iv clinical trial, and the authors published two papers already without showing the critical data, the therapeutic effects of EE. In Table 2, the authors showed that HDL-C levels were significantly decreased after 16-week treatments of E or EE, but difference is very small (36 vs. 39 for E, 40 vs. 42 for EE). Even if this small difference is statistically significant, the authors should provide clinical data and evidence to support the effects of E and EE. How only 2-3 mg/dL differences in HDL-C affect in conditions of diabetes? Description of NCT03932721 says that the authors will analyze levels of glucose and HbA1c, but no data are provided for these parameters. Conditions of diabetes were improved in E and EE groups or not? How were decreased HDL-C levels in E and EE groups associated with conditions of diabetes? The authors should answer these questions. Data provided in this manuscript showed the effects of E and EE on HDL-C levels, but it is unclear if these decreased HDL-C is associated with therapeutic effects of E or EE or conditions of diabetes. This study is not a basic study, but a clinical trial. It is inappropriate not to show clinical data and show something different, which may be or may not be associated with diseased conditions. Readers expect clinical data for clinical trials and hence I will not accept this manuscript unless the authors provide clinical data and evidence for the effects of E and EE. Clinical trials often show disappointing results. The authors should show all clinical data and conclude that E or EE did not provide clinical improvements or significant differences if results were not favorable.
Author Response
In this study, Bonilha et al. analyzed data of the EXCEED-BHS3 trial to seek the therapeutic effects of the combination of empagliflozin and evolocumab in patients with type 2 diabetes. This clinical trial is registered as NCT03932721, and the authors published a paper to show its concepts and methods (ref#19). The authors published another paper to show that plasma levels of nitrate and nitrite were increased in EE group, compared to E group (ref#11). In this manuscript, the authors showed that serum levels of HDL-C were significantly elevated in both E and EE groups, and the effects of EE were greater than E only. However, NCT03932721 is a phase iv clinical trial, and the authors published two papers already without showing the critical data, the therapeutic effects of EE. In Table 2, the authors showed that HDL-C levels were significantly decreased after 16-week treatments of E or EE, but difference is very small (36 vs. 39 for E, 40 vs. 42 for EE). Even if this small difference is statistically significant, the authors should provide clinical data and evidence to support the effects of E and EE. How only 2-3 mg/dL differences in HDL-C affect in conditions of diabetes? Description of NCT03932721 says that the authors will analyze levels of glucose and HbA1c, but no data are provided for these parameters. Conditions of diabetes were improved in E and EE groups or not? How were decreased HDL-C levels in E and EE groups associated with conditions of diabetes? The authors should answer these questions. Data provided in this manuscript showed the effects of E and EE on HDL-C levels, but it is unclear if these decreased HDL-C is associated with therapeutic effects of E or EE or conditions of diabetes. This study is not a basic study, but a clinical trial. It is inappropriate not to show clinical data and show something different, which may be or may not be associated with diseased conditions. Readers expect clinical data for clinical trials and hence I will not accept this manuscript unless the authors provide clinical data and evidence for the effects of E and EE. Clinical trials often show disappointing results. The authors should show all clinical data and conclude that E or EE did not provide clinical improvements or significant differences if results were not favorable.  
Response from authors: We thank the reviewer for the contribution. In the publication (ref#11) we showed that in individuals with TDM2 undergoing treatment with SGLT2i, the addition of evolocumab improves endothelium-mediated arterial dilation (assessed by FMD) and nitric oxide production both under resting conditions and after stimulation with ischemia and reperfusion. In this article, although our primary endpoint was the assessment of endothelial function through FMD, we showed that after 16 weeks of treatment, changes in 24-hour systolic and diastolic BP were equivalent between groups. There was a similar decrease in BMI in both groups. The median reduction in HbA1c and fasting blood glucose were also equivalent between the groups. However, to make it clearer, we created a supplementary table with these clinical data showing that diabetes conditions improved in both groups. We add clinical data, from line 77 to 80, as follows: “After 16 weeks of treatment, changes in systolic BP, diastolic BP and mean reduction in HbA1c and fasting glucose were equivalent between groups as seen in the supplementary table”.
The reviewer comments that in table 2 we show that HDL-C levels decreased significantly after 16 weeks of treatment with E or EE, however, HDL-C levels increased in both favoring smaller diameter particle, such as 2a, 3a, 3b and 3c in both therapies.
Even though there is a difference of 2-3 mg/dL in HDL-C concentration, these are important results. Gordon et al. (10.1161/01.CIR.79.1.8), estimated that a 1 mg/dL increase in HDL was associated with a significant 2–3% reduction in the risk of heart disease. Since diabetic patients have an increased risk of developing cardiovascular disease, this change can be beneficial.
Round 2
Reviewer 5 Report
Comments and Suggestions for Authors
In this study, Bonilha et al. analyzed data of the EXCEED-BHS3 trial to seek the therapeutic effects of the combination of empagliflozin and evolocumab in patients with type 2 diabetes.
In this revised manuscript, the authors addressed my comments, but I still feel that this study is misleading.
This is a clinical trial, not a basic study. The authors show results, limitations, and conclusions in clinical aspects clearly. Clinical data should be shown as main data, not supplementary. The authors claim as if the treatments were effective, but there is no clinical evidence to support this claim. The authors should conclude clearly that this treatment did not provide clinical benefits in this cohort, otherwise this study is misleading. The authors claim that diabetes conditions were improved, but not enough evidence supports this. Decreased levels of HDL-C are minimal, and it is unclear if this decrease is associated with improved conditions (clinical data do not support so). The authors say that 2-3 mg/dL difference is important, but no evidence is provided to prove this. The study from Gordon et al. was not for diabetes. It is well known that HDL-C is a good cholesterol so it is not surprising. My point is that the authors should show the correlation between 2-3 mg/dL increase and clinical benefits in this trial. Clinical data show no difference between groups, so it means that this treatment does not provide any benefit. The authors should say this clearly or this manuscript cannot be accepted because otherwise this clinical trial is misleading.
Author Response
This is a clinical trial, not a basic study. The authors show results, limitations, and conclusions in clinical aspects clearly. Clinical data should be shown as main data, not supplementary. The authors claim as if the treatments were effective, but there is no clinical evidence to support this claim. The authors should conclude clearly that this treatment did not provide clinical benefits in this cohort, otherwise this study is misleading. The authors claim that diabetes conditions were improved, but not enough evidence supports this.
Response from authors: Thank you for your valuable feedback. The data on systolic and diastolic blood pressure, fasting blood glucose and glycated hemoglobin that were in supplementary material are now in the main text, highlighted as table 2 and in red. The EXCEED study is indeed a translational clinical trial, but it primarily focuses on surrogate outcomes related to the distribution of HDL subspecies. We enhanced the description of this aspect in the methods section to ensure that readers do not have unrealistic expectations regarding the study endpoint assessments. We also removed any inferences regarding the clinical benefit that may result from this effect on HDL metabolism and mentioned the actual nature of this translational phase 2 trial.
Decreased levels of HDL-C are minimal, and it is unclear if this decrease is associated with improved conditions (clinical data do not support so). The authors say that 2-3 mg/dL difference is important, but no evidence is provided to prove this. The study from Gordon et al. was not for diabetes. It is well known that HDL-C is a good cholesterol so it is not surprising. My point is that the authors should show the correlation between 2-3 mg/dL increase and clinical benefits in this trial.
Response from authors: Thank you for your follow-up inquiry. As previously mentioned, the primary endpoint of our study was the change in the proportion of HDL subspecies. The study duration and sample size were determined based on this specific objective. Consequently, we did not capture clinical events that could be compared with changes in HDL levels. This kind of study are necessary to generate hypothesis that would or not deserve to be tested on a large and long-lasting clinical trial based on hard endpoints.
Clinical data show no difference between groups, so it means that this treatment does not provide any benefit. The authors should say this clearly or this manuscript cannot be accepted because otherwise this clinical trial is misleading.
Response from authors: We thank the reviewer for the helpful suggestion. We added the comment on this limitation in order to prevent misleading the readers.
Round 3
Reviewer 5 Report
Comments and Suggestions for Authors
No further comments